# Optimal Treatment with Cannabis Extracts Formulations Is Gained via Knowledge of Their Terpene Content and via Enrichment with Specifically Selected Monoterpenes and Monoterpenoids

**DOI:** 10.3390/molecules27206920

**Published:** 2022-10-15

**Authors:** Noa Raz, Aharon M. Eyal, Elyad M. Davidson

**Affiliations:** 1Bazelet Medical Cannabis Group, Or Akiva 3065101, Israel; 2Department of Anesthesiology, CCM and Pain Relief, Hadassah Hebrew University Hospital, Jerusalem 9112001, Israel

**Keywords:** terpenes, monoterpenes, cannabis extracts, full-spectrum, terpenes enrichment, entourage effect

## Abstract

Differences between therapeutic effects of medical cannabis inflorescences and those of their extracts are generally attributed to the differences in administration form and in the resultant pharmacokinetics. We hypothesized that difference may further extend to the composition of the actually consumed drug. Cannabinoid and terpene contents were compared between commercial cannabis inflorescences (*n* = 19) and decarboxylated extracts (*n* = 12), and between inflorescences and decarboxylated extracts produced from them (*n* = 10). While cannabinoid content was preserved in the extracts, a significant loss of terpenes was evident, mainly in the more volatile monoterpenes and monoterpenoids (representing a loss of about 90%). This loss changes the total terpene content, the proportion of monoterpenes out of the total terpenes, and the monoterpene/cannabinoid ratio. Terpene deficiency might impair extracts’ pharmacological efficacy and might contribute to the patients’ preference to inflorescences-smoking. This argues against the validity of terms such as “whole plant” and “full spectrum” extracts and creates a misleading assumption that extracts represent the pharmacological profile of the sourced inflorescences. Furthermore, it reduces the diversity in extracts, such as loss of differences between sativa-type and indica-type. Enriching cannabis extracts with selected terpenes may provide a suitable solution, generating a safe, precise, and reproducible drug with tailored cannabinoid and terpene contents. Careful selection of terpenes to be added enables tailor-made extracts, adjusted for various medicinal aims and for different populations.

## 1. Introduction

The number of medical cannabis users is increasing rapidly. Alongside this, the number and variety of authorized medical cannabis compositions and delivery methods available on the market continuously increases. Physicians and caregivers prescribing medical cannabis, as well as patients who use it regularly, face difficulties in selecting the most suitable preparation. While there might not be sufficient scientific data to support such selection, it is clear that the main objectives for the most suitable products include (i) an optimal composition (an optimal combination of active pharmaceutical ingredients (APIs)) for the specifically treated indication; (ii) precision—an exact knowledge and control of the administered dose of each important API; (iii) reproducibility—as-perfect-as-possible repeatability in the composition of the selected product, week after week and month after month; (iv) ability to adjust the composition for varying requirements, e.g., fitting to changes in day activity; and (v) ability to adjust the composition to varying population needs, for example women vs. men, and children vs. aging population.

A large fraction of the medical cannabis users smoke their cannabis drug (in Israel, more than 80% [1]). This preference seems unfortunate. Firstly, smoking is accompanied by known health-threatening issues, including the inhalation of ash particles and of toxic oxidation products. This is even worse when smoking cannabis mixed with tobacco, which is a common practice among many medical cannabis patients. Secondly, the short-onset time in smoking (beneficial in acute cases) is less important for most medical cannabis users, who suffer from chronic ailments, and who can benefit from the extended effect of the oils. Thirdly, smoking suffers badly from inaccurate analysis and poor reproducibility [2,3] making it unsuitable for a precise medicinal treatment. Impaired accuracy and reproducibility in smoking results from multiple reasons, including agricultural effects and post-harvest and industrial processing parameters, as well as from non-standardized consumption factors [4]. A seemingly preferred delivery form is based on cannabis extracts, also referred to as cannabis oils, which are produced by extracting cannabis inflorescences and diluting in a vegetable oil. Extracts are consumed sublingually, via spray into the oral cavity, or further formulated into tablets and capsules. Each one of these forms has its own pharmacokinetics and absorption rate. Compared to smoking, extracts-based products are safer for use, provide longer lasting effects, and enable better accuracy and reproducibility. Additionally, the majority of known clinical trials are based on administering extracts, rather than on smoking, e.g., [5,6,7,8,9].

The present study is directed at identifying a potential drawback of cannabis oils (in addition to undesired flavor in some formulations), and at potential means for resolving that drawback. It focuses on significant differences in the compositions of the consumed oils compared with those of the corresponding smoked inflorescence, differences which might reduce the efficacy of the former in some treatment cases. It suggests that, contrary to expectation by most caregivers and users, the content and composition of the APIs in the extracts differ markedly from those in the inflorescence used to produce them. The main difference is not in the cannabinoid content, but rather in the content of the terpenes, which are not reported in most commercially marketed cannabis products and in many clinical trials. To evaluate possible differences, the compositions and amounts of cannabinoids and terpenes in cannabis inflorescences and extracts, including inflorescences and decarboxylated extracts produced from them, were compared.

More than 200 different terpenes and terpenoids [10] have been identified in cannabis, out of which about 20 are most common, including myrcene, limonene, pinene, linalool, terpinolene, β-caryophyllene, and humulene. Most cannabis terpenes belong to one of the following groups, arranged here according to increasing boiling point order: monoterpenes (including myrcene, pinene, limonene, ocimene, and terpinolene), monoterpenoids (including linalool, terpineol, and geraniol), sesquiterpenes (including caryophyllene and humulene), and sesquiterpenoids (including nerolidol, guaiol, and bisabolol) [10,11,12,13], see Table 1. The total terpene concentration in cannabis inflorescences was commonly in the 1% range, however, due to selective breeding, this concentration has risen to about 3% in some inflorescences [11], reaching terpene-cannabinoid ratios of up to about 1:10. Despite these relatively low concentrations, it is argued that terpenes modulate the pharmacodynamic effects in cannabis, e.g., [14,15,16,17,18,19], as further discussed below.

Terpenes have been suggested, over thousands of years, to have various, broadranging therapeutic properties, which provide the basis for traditional and modern aromatherapy and herbal medicine. Different terpenes have been described as having various therapeutic properties, for instance, myrcene: analgesic, anti-inflammatory, sedative, and muscle relaxant; α-pinene: anti-oxidant, anti-inflammatory, and bronchodilator; linalool: anti-convulsant, analgesic, sedative and anti-depressant; limonene: ameliorates stress and depression; and β-caryophyllene: gastroprotective, analgesic, and antibiotic effect [15,17,20,21]. Within the endocannabinoid system, terpenes were demonstrated to activate or modulate cannabinoid receptor type 1 (CB1), cannabinoid receptor type 2 (CB2), transient receptor potential ankyrin 1 (TRPA1), transient receptor potential vanilloid 1 (TRPV1) and peroxisome proliferator-activated receptor (PPAR) receptors [18,22,23,24]. In addition to their own therapeutic properties, terpenes were suggested to affect the therapeutic activity of cannabinoids via synergic and/or entourage effects [25]. The role of terpenes in the entourage effect of cannabis was first suggested by Russo [15], and was subsequently supported by studies demonstrating the role of specific terpenes and their interaction with cannabinoids in increasing tetrahydrocannabinol (THC) antinociception [18], THC’s anxiolytic effect [19], THC and CBD (cannabinol) cytotoxic activity [16], and more [11,14,21,26,27,28].

## 2. Results and Discussion

Although the role of terpenes in cannabis’ pharmaceutical effect is gaining increased acceptance [11,15,16,17,18,19,21,28,29,30], most scientific reports and commercial products are provided with information regarding their cannabinoid content, but with no terpene data. Where terpene content is presented, it is typically in the form of concentration or in the form of the fraction of each terpene out of the total terpene content [12,27]. However, for medical applications and for the discussion of the entourage/synergistic effect, more important are the administered terpene doses and the relative proportion of terpenes to cannabinoids (See also [31]).

Cannabinoid and terpene data for inflorescences of nineteen different cannabis chemovars (or chemotypes, frequently related as “cannabis strains”) grown in Israel and marketed as medical cannabis inflorescences are presented as concentrations and as administered doses, i.e., as milligrams of each terpene per 100 mg total cannabinoid content (Table 2 and Figure 1).

Terpenes are arranged according to their boiling points. Total terpene contents in the cannabis inflorescences tested here range between 0.4% and 2.3%, and terpene–cannabinoid ratios range between 2.1 and 12.2 mg total terpenes per 100 mg total cannabinoids (Table 2). It is important to note that the inflorescences are rich in monoterpenes and monoterpenoids, forming together 46% to 83% of the total terpenes in the composition and up to 10.1 mg per 100 mg cannabinoids. The most prevalent monoterpenes are myrcene, alpha pinene, beta pinene, and limonene, reaching up to 4 mg each per 100 mg cannabinoids. These results are in agreement with prior reports [10,11,12,16]. Another important aspect to keep in mind is that, as shown in Figure 1, the diversity of the inflorescences is derived, to a large extent, from the monoterpene and monoterpenoid content.

On average, medical cannabis patients in Israel consume about 1 g of inflorescence per day [1], containing up to about 200 mg cannabinoids. The corresponding total terpene content ranges from about 4 to 24 mg. Note that these figures are true for the content of the consumed cigarettes, but not necessarily for the doses of actually consumed cannabinoids and terpenes, since active components are lost during smoking (via burning or evaporation prior to absorption within the airways [4]). 

Cannabinoid and terpene data for commercial medical cannabis extracts (oils) in the Israeli market are shown in Table 3 and in Figure 2. In these oils, the total terpene contents (up to 0.93%) and the content of total terpenes per 100 mg total cannabinoids (up to 4.3 mg, Table 3) are low compared with those in the inflorescences. In oils, the combined contents of sesquiterpenes and sesquiterpenoids are similar to those in the inflorescences (up to 3.8 vs. up to 4.3 mg per 100 mg cannabinoids), but those of the monoterpene and monoterpenoids, combined, is notably lower (up to 0.55 mg vs. 10.1 mg). As a result, the proportions of monoterpenes and monoterpenoids, combined, drop from up to 83% in the inflorescence to up to 22% in the oils. 

Comparisons between the inflorescence and oil samples are provided in Figure 3 [Total terpene content- inflorescences: M = 7.6, SD = 2.6; oils: M = 2.8, SD = 1.0 mg/ 100 mg cannabinoids, (t(25.768) = 7.3, *p* < 0.001, d = 2.3). Total monoterpene and monoterpenoid content: inflorescences: M = 5.0, SD = 2.1; oils: M = 0.3, SD = 0.1 mg/ 100 mg cannabinoids, (t(18.208) = 9.6, *p* < 0.001, d = 2.8). Total sesquiterpene and sesquiterpenoid content: inflorescences: M = 2.6, SD = 0.9); oils: (M = 2.4, SD = 0.9) mg/ 100 mg cannabinoids, (t(29) = 0.5, *p* > 0.05, d = 0.2). Independent sample *t*-test]. It is important to note that this marked reduction in the fraction of the monoterpenes nearly eliminates the diversity in the oils with regards to the terpenes profile, as seen in Figure 2.

The lower monoterpene and monoterpenoid content of the oils was previously reported [12,32,33] and is typically attributed to their relative volatility, leading to evaporation during the extraction and during the thermal treatment steps of typical cannabis oil manufacturing processes. Those steps include inflorescences’ drying, grinding, extraction, and decarboxylating [4,33] the natural acid tetrahydrocannabinolic acid (THCA) and cannabidiol acid (CBDA) into their neutral THC and CBD form (the endocannabinoids receptors’ agonists form, [34,35,36,37]).

Since the oils in Figure 2 do not directly correspond to the inflorescences in Figure 1, and in order to further verify the selective loss of monoterpenes and monoterpenoids during cannabis oil production, a dedicated industrial trial was conducted, comparing the terpene contents of ten inflorescences to those of concentrated decarboxylated extracts produced from these inflorescences. The results, presented in Table 4 and in Figure 4, confirm the previous observations: total terpene content drops due to a drastic drop in the content of monoterpenes. Since there is no significant reduction in the content of the sesquiterpene and sesquiterpenoids, the fraction of monoterpenes out of the total terpene content is markedly reduced in the extracts. [Total terpene content: inflorescences: M = 11.2, SD = 0.4; oils: M = 5.7, SD = 0.3 mg/100 mg cannabinoids, ((t(18) = 2.9, *p* < 0.01, d = 1.3). Total monoterpene and monoterpenoid content: inflorescences: M = 5.2, SD = 0.5; oils: M = 1.1, SD = 0.1 mg/100 mg cannabinoids, (t(10.646) = 10.0, *p* < 0.001, d = 4.5). Total sesquiterpene and sesquiterpenoid content: inflorescences: M = 5.9, SD = 0.4; oils: M = 4.6, SD = 0.3 mg/100 mg cannabinoids, (t(18) = 0.414, *p* > 0.05, d = 0.185). Independent sample *t*-test]. See Appendix A for further information).

To summarize these results, comparing commercial cannabis inflorescences to commercial cannabis oils consumed by medical cannabis patients in Israel, a significantly lower total amount of terpenes was found in the latter and a much-reduced fraction of the monoterpenes and monoterpenoids out of the total terpene content. The same observations were found when directly comparing inflorescences to decarboxylated extracts produced from them. Future studies should further address potential differences in bioactive components, other than terpenes, such as flavonoids.

This deficiency in terpenes of the oils, particularly in that of monoterpenes and monoterpenoids, may have major implications. This is since terpenes have therapeutic properties of their own and, furthermore, may modify the therapeutic activity of cannabinoids via a synergic and/or entourage effect [25], as discussed above.

Of particular importance here is the therapeutic role of monoterpenes, which form a major fraction of the terpenes in inflorescences, but are mostly lost in the oils manufactured from them [10,11,12,38]. As shown above, monoterpenes are also mainly responsible for the diversity of the chemovars. Previous reports have suggested that monoterpenes have a primary role in classifying a chemovar as having a dominant indica effect vs. a dominant sativa effect. The presence of myrcene has the strongest association with the Indica type, providing a sedative, effect, while chemotypes with low myrcene levels have a more energetic type [10,26,27]. The presence of terpinolene, another monoterpene, was found by Casano et al. [26] to be mostly associated with the sativa type.

Given the deficiency of oils in these important active pharmaceutical ingredients, there is no support for terms such as “whole plant extracts” or “full spectrum extracts”, when applied to cannabis extracts/oils. Most studies, clinical trials, and reviews using these terms, e.g., [39,40,41,42], lack important composition information and are not really conclusive without terpene content data. Furthermore, these terms are misleading, creating the expectation that these “whole plant/full spectrum” extracts represent the composition of the original plant. Similarly deficient are clinical trials of oils specified by the strain used to form them, assuming that this strain’s properties are retained in the formed extract [43,44,45]. As the composition of terpenes in extracts varies with cultivation conditions and depends on the extraction and thermal process steps used [4,33,42,46,47], no reproducibility is guaranteed for different extracts of the same plant source, unless directly analyzed and corrected. It is worth noting that Sativex^®^ and Cannador^®^, the intensively studied and used cannabis extracts, are defined by their THC and CBD content, with no data on the other extracted ingredients [40,48]. The deficiency, or major reduction in monoterpene content in the oils, may support the hypothesis regarding the impaired efficacy of oils compared with smoking, which drives patients to prefer the latter. It also points out the directions to resolve this issue and to make oils the preferred formulations, as further detailed below.

## 3. Conclusions and Implications

In conclusion, while smoking has many disadvantages, there could be some justifications to patients’ claims indicating an inferior effect of the oils. Oils significantly lack in monoterpene content as compared with inflorescences, the relative proportion of the various terpenes changes, and the differentiation gained by the content of specific monoterpenes is lost. As an example, monoterpenes are responsible for the differences between the effects of sativa and indica type chemovars, but those distinguishing terpenes are lost in oil production, so that oils produced from sativa-type chemovars do not differ much from oils produced from indica-type chemovars of similar cannabinoid content.

Some publications suggest collecting the lost terpenes and adding them back to the produced oil [33,47]. Given the number of steps wherein terpenes evaporate and given the need to separate terpene vapors from other gaseous components (e.g., solvent vapors), such collection is practically infeasible in industrial production, processing dozens of kilograms of inflorescence per batch. However, there is a much simpler solution. The terpenes of interest are produced not only in cannabis and could be obtained from other sources, e.g., hops (myrcene), pine (pinene), and lavender (linalool). Those terpenes are commercially available, most times at high purity and at a relatively low price. Accordingly, terpene loss may be resolved by obtaining those terpenes from other plants and adding them to the generated cannabis extracts. This modified method has several important advantages, as detailed below.

First are accuracy and reproducibility. According to the modified method, the extracts formed are analyzed for their terpene content and then calculated amounts of each desired terpene are added to accurately reach the selected level. This way, the final concentration of each terpene would be the same in each preparation, independent of the extracted inflorescence or of the production parameters. Additionally, purified cannabinoid (“isolates”) may be used without worrying about the terpenes lost in the purification step [49]. Similarly, synthetically or bio-synthetically produced cannabinoids formed with no terpenes may be used for desired compositions.

An even more important advantage is, that one is not limited to terpene compositions replicating those of particular chemovars. It should be kept in mind that the terpene compositions of a “whole plant” or a “full spectrum” are not designed by nature for the benefit of the medical user, but rather for the needs of the cannabis plant, such as plant protection [12,50,51] (but see also Namdar et al. [16] with a different reasoning). Additionally, generating new compositions for improved efficacy and for new applications does not require years of costly breeding of new chemovars. Cannabinoids and terpenes can be simply added (or removed) to reach any desired composition. For example, the loss of the sativa/indica nature in extraction could be resolved by adding terpenes for recreating the original plant terpene composition. However, why stop there? Terpenes known to improve sleep quality can be added to various oils for night use, while terpenes improving awareness and functionality can be added to various oils designed for day use. Going even further, cannabis oil compositions can specifically be designed for various medical functions, e.g., pain management or anxiety relief. Suitable terpene compositions, based on terpenes’ pharmaceutical properties and their role in modulating the cannabinoid pharmaceutical effects [15,16,17,18,19,21], will be added to each designated product. Adjusting the composition to varying requirements (e.g., pain composition for day use vs. ones for night use) or to varying populations (e.g., children vs. adults or men vs. women) allows further tailoring the cannabis oils medicinal product to desired needs. This enrichment of oils with selected terpenes for selected needs is robustly doable in industrial settings and is already implemented in products available on Israeli market [52,53,54,55].

## 4. Materials and Methods

### 4.1. Materials

Commercial medical cannabis extracts (“cannabis oils”) and commercial inflorescences, produced by various Israeli licensed producers, were obtained from Bazelet Pharma (Or Akiva, Israel). Cannabinoid standards: cannabinol (CBN), cannabichromene (CBC), cannabichromenic acid (CBCA), cannabigerolic acid (CBGA), cannabigerol (CBG), cannabidiolic acid (CBDA), cannabidiol (CBD), delta-9-tetrahydrocannabinoic acid (THCA) and delta-9-tetrahydrocannabinol (THC) were purchased from Cerrilliant (Cerilliant Corporation, Round Rock, TX, USA). Cannabidivarin (CBDV), cannabidivarinic Acid (CBDVA), tetrahydrocannabivarin (THCV), and delta-8-tetrahydrocannabinol (Δ8-THC) were purchased from Restek (Bellefonte, PA, USA). All cannabinoid standards are Certified Reference Materials (CRM) standards, at 1000 µg/mL in methanol. Terpene standards were purchased from Restek (Bellefonte, PA, USA), Merck (Rosh-Ha’ayin, Israel), and Phyto Lab (Vestenbergsgreuth, Germany) (See Appendix A for further details). Ethanol for standard solutions and samples preparation was HPLC grade (J.T. Baker, Phillipsburg, NJ, USA). All terpene standards are Certified Reference Materials (CRM) standards, at 2500 µg/mL in isopropanol.

### 4.2. Methods

In order to directly evaluate the terpene content changes during a typical industrial cannabis oil production, an industrial trial was performed in which inflorescences of ten different chemovars were analyzed and then extracted. Inflorescences of each chemovar went through the following steps. A 10 kg batch was dried at ambient temperature and at 50% relative humidity to reach a moisture content of 11–12%. The 10 kg of dried inflorescences were ground and then contacted at ambient temperature with 60 L ethanol, while mixing for a duration of 30 min. The formed ethanol-diluted extract was separated. The same extraction procedure was repeated for a second time with 40 more liters of fresh ethanol and the second formed ethanol-diluted extract was separated from the residual extracted plant material. Multiple previous tests have confirmed full cannabinoids extraction in this procedure. The two ethanol-diluted extracts were combined and the ethanol was evaporated out at 40 °C under reduced pressure, to form a concentrated extract (of about 60% THCA). This concentrated extract was held at 110 °C during an hour, for decarboxylation of the acid form THCA to THC. The same procedure was repeated for each one of the ten chemovars. Analysis and extract production were conducted at the IGMP Bazelet Medical Cannabis manufacturing plant in Or Akiva, Israel.

### 4.3. Analysis

Inflorescences and extracts were analyzed for their cannabinoid and terpene content, using high performance liquid chromatography (HPLC) and gas chromatography (GC), respectively. Inflorescences were prepared for analysis via a standardized procedure, comprising extraction of 0.5 gr inflorescences in 45 gr ethanol.

High Performance Liquid Chromatography (HPLC): The analysis of cannabinoids was carried out on HPLC Waters PDA 2996 (Waters Corporation, Milford, MA, USA), equipped with a pump, autosampler, column-oven, and a Photodiode Array detector (PDA) detector. The analytical balance was Mettler Toledo MS205DU (Mettler Toledo, Columbus, OH, USA). The HPLC column used was Phenomenex Luna Omega C18 column (Phenomenex, Torrance, CA, USA). The mobile phase was buffer (ammonium acetate): acetonitrile, at 1:1 ratio, at a constant flow of 0.1 mL/min. Detection used wavelength of 220 nm, injected volume: 10 µL. The method is fully validated for 12 cannabinoids in line with the requirements of the International Council for Harmonisation of Technical Requirements for Pharmaceuticals for Human Use (ICH) [56] guidelines, Israeli Medical Cannabis Association (IMCA), European Pharmacopoeia (EP) [57], and United States Pharmacopeia (USP) [58]. The nominal working concertation is 100 µg/mL and the method range is 0.1–120.0% of that nominal working concentration, proved by linearity, precision, and accuracy studies. The limit of detection of the method is 0.1 µg/mL and the limit of quantitation −0.2 µg/mL. Uncertainties were within 5% of the reported value. Total cannabinoid content was calculated as if all of the cannabinoids were in their decarboxylated form (See also Appendix A).

Gas Chromatography (GC): Terpene analysis was carried out on Agilent Technologies GC system model 6890 N (Agilent Technologies, Santa Clara, CA, USA) equipped with Flame Ionization Detector (FID). Identification was based on the retention times of the Certified Reference Materials (CRM) standards and was verified by GC MS (Gas chromatography–mass spectrometry) at Aminocann (Aminolab, Ness Ziona, Israel). A CTC autosampler (Pal RTC, CTC analytics, Zwingen, Switzerland) was used. The column used was Phenomenex ZB-624plus (Phenomenex, Torrance, CA, USA) with helium as carrier at 1.2 mL/min constant flow. The method is fully validated for 25 terpenes likely to be present in cannabis. The method is fully validated according to the requirements of the International Council for Harmonisation of Technical Requirements for Pharmaceuticals for Human Use (ICH) [56] guidelines, Israeli Medical Cannabis Association (IMCA), European Pharmacopoeia (EP) [57], and United States Pharmacopeia (USP) [58]. The range of the method was between 200–4000 µg/mL, proved by linearity, precision, and accuracy studies. The limit of reporting was 200 µg/mL. Uncertainties were within 5% of the reported value. Terpenes lacking analytical standards are presented by their retention time and their content was estimated by calculating their area according to α-Humulene response factor. Retention times for identified terpenes are also provided (see Table 4) for reference.

Statistical analyses were performed using SPSS 20.0 (IBM Corp., Armonk, NY, USA) (See Appendix A for further detail).

## Figures and Tables

**Figure 1 molecules-27-06920-f001:**
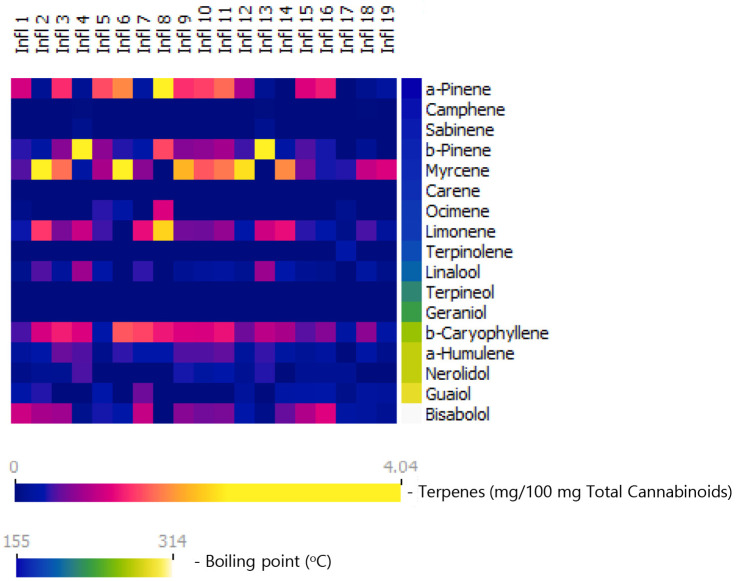
Terpene content in nineteen commercial cannabis inflorescences marketed in Israel, presented as milligrams of each terpene per 100 mg total cannabinoid content. Terpenes are arranged according to their boiling points. As seen, monoterpenes comprise the largest terpene group in these inflorescences. Furthermore, the diversity between inflorescences is driven, to a large extent by monoterpenes.

**Figure 2 molecules-27-06920-f002:**
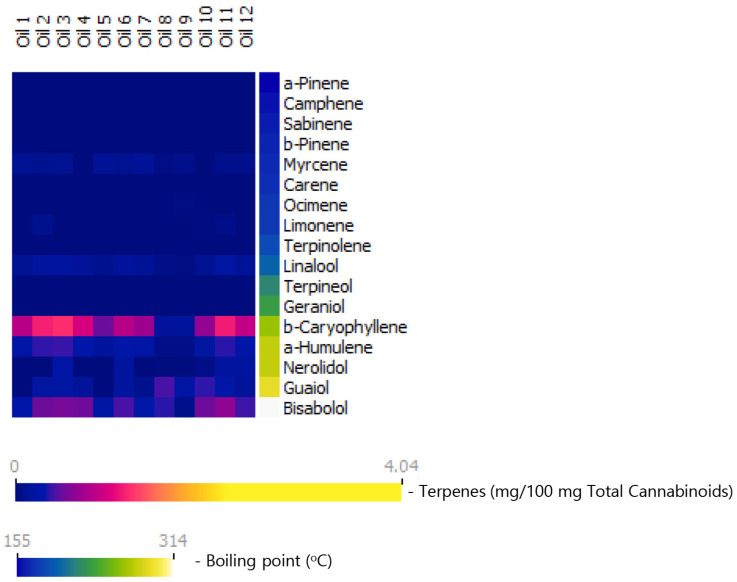
Terpene content in commercial olive-oil-diluted cannabis extracts (cannabis oils) marketed in Israel, presented as milligrams of each terpene per 100 mg total cannabinoid content. Terpenes are arranged according to their boiling points. As seen, monoterpenes and monoterpenoids are majorly lost in these extracts. Monoterpene loss nearly eliminates the diversity in extracts with regard to terpene profile.

**Figure 3 molecules-27-06920-f003:**
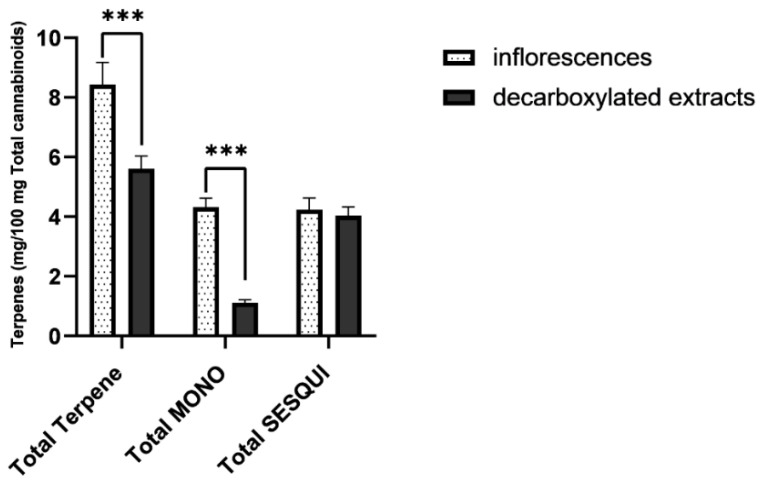
Averaged terpene/cannabinoid proportions of commercial cannabis inflorescences (*n* = 19) and commercial cannabis extracts (*n* = 12), presented as milligrams of each terpene per 100 mg total cannabinoid content. As seen, total terpene proportion in cannabis extracts is significantly reduced. This reduction derives from a significant loss of monoterpene and monoterpenoids. Total MONO = total monoterpenes and monoterpenoids. Total SESQUI = total sesquiterpenes and sesquiterpenoids. Asterisks denote significance level. *** *p* < 0.001. Error Bars denote SEM.

**Figure 4 molecules-27-06920-f004:**
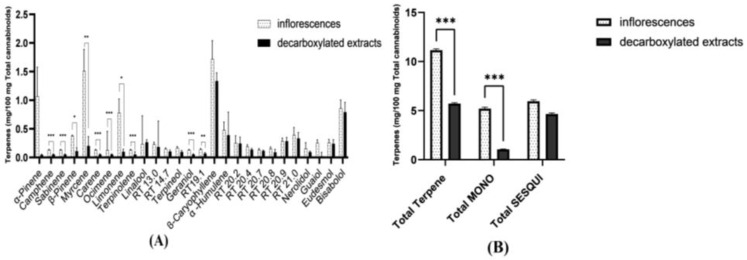
Averaged terpene content of ten inflorescences and of the decarboxylated extracts produced from them, presented as milligrams of each terpene per 100 mg total cannabinoids content. (**A**) Averaged data per each terpene. (**B**) Averaged data for total terpene content, total monoterpene and monoterpenoid content (total MONO), and total sesquiterpene and sesquiterpenoid content (total SESQUI) (see Appendix A for further details). As seen, while the content of sesquiterpenes and sesquiterpenoids is largely kept in the formed extracts, monoterpenes are completely or mostly lost. Asterisks denote significance level. * *p* < 0.05, ** *p* < 0.01, *** *p* < 0.001. Error Bars denote SEM.

**Table 1 molecules-27-06920-t001:** Terpenes’ classification.

Terpene Class	Terpenes	Boiling Points (°C)
**Monoterpenes (C_10_H_16_)**	α-Pinene	155
	Camphene	159
	Sabinene	163
	β-Pinene	166
	Myrcene	168
	Carene	171
	Ocimene	175
	Limonene	176
	Terpinolene	185
**Monoterpenoids (C_10_H_18_O)**	Linalool	198
	Terpineol	217
	Geraniol	230
**Sesquiterpenes (C_15_H_24_)**	β-Caryophyllene	263
	Humulene	276
**Sesquiterpenoids (C_15_H_26_O)**	Nerolidol	276
	Guaiol	290
	Bisabolol	314

**Table 2 molecules-27-06920-t002:** Cannabinoid and terpene compositions in commercial medical cannabis inflorescences in Israel. Data is presented as absolute percentage ^1^ and in the form of milligrams of each terpene per 100 mg total cannabinoid content (*in italic*).

Compound Name	Inf 1	Inf 2	Inf 3	Inf 4	Inf 5	Inf 6	Inf 7	Inf 8	Inf 9	Inf 10	Inf 11	Inf 12	Inf 13	Inf 14	Inf 15	Inf 16	Inf 17	Inf 18	Inf 19
THCA	10.7	14.5	18.8	21.4	22.8	24.4	17.9	13.1	20.8	21.9	19.4	19.2	19.7	16.5	14.5	16.1	20.4	7.0	0.6
THC	3.5	1.6	1.7	1.3	0.7	1.0	1.3	1.0	1.7	0.6	1.9	1.3	0.8	1.1	0.6	3.8	0.9	0.9	0
Total THC	12.9	14.3	18.2	20.1	20.7	22.4	17.1	12.5	20.0	19.8	18.9	18.1	18.1	15.6	13.3	17.9	18.8	7.0	0.5
CBDA																	0.1	12	16.8
CBD																		1.1	0.7
Total CBD	<0.1	<0.1	<0.1	<0.1	<0.1	<0.1	<0.1	<0.1	<0.1	<0.1	<0.1	<0.1	<0.1	<0.1	<0.1	<0.1	0.1	11.6	15.4
Total quantified Cannabinoids	12.9	14.3	18.2	20.1	20.7	22.4	17.1	12.5	20.0	19.8	18.9	18.1	18.1	15.6	13.3	17.9	18.9	18.7	15.9
α-Pinene	0.18 *1.39*	0.03 *0.19*	0.31 *1.71*	0.03 *0.17*	0.41 *1.94*	0.52 *2.31*	0.05 *0.30*	0.48 *3.87*	0.36 *1.76*	0.37 *1.86*	0.40 *2.12*	0.20 *1.10*	0.03 *0.17*		0.20 *1.46*	0.29 *1.61*		0.03 *0.15*	0.04 *0.28*
Camphene				0.01 *0.05*									0.01 *0.05*					0.01 *0.03*	
Sabinene				0.03 *0.13*									0.03 *0.14*						
β-Pinene	0.07 *0.52*	0.04 *0.31*	0.16 *0.87*	0.67 *3.33*	0.19 *0.89*	0.11 *0.5*	0.07 *0.38*	0.24 *1.91*	0.18 *0.87*	0.18 *0.91*	0.20 *1.04*	0.10 *0.56*	0.60 *3.31*	0.05 *0.35*	0.08 *0.62*	0.09 *0.48*		0.04 *0.21*	
Myrcene	0.08 *0.64*	0.49 *3.44*	0.39 *2.13*	0.07 *0.32*	0.23 *1.08*	0.91 *4.04*	0.15 *0.87*		0.54 *2.66*	0.39 *2.00*	0.42 *2.21*	0.55 *3.06*		0.36 *2.33*	0.10 *0.78*	0.09 *0.49*	0.10 *0.51*	0.25 *1.35*	0.23 *1.47*
Carene																			
Ocimene	0.01 *0.09*				0.11 *0.51*	0.08 *0.36*		0.18 *1.42*									0.03 *0.15*		
Limonene	0.06 *0.47*	0.25 *1.76*	0.14 *0.78*	0.26 *1.30*	0.12 *0.57*		0.26 *1.53*	0.37 *2.93*	0.15 *0.76*	0.15 *0.75*	0.18 *0.95*	0.08 *0.45*	0.24 *1.34*	0.24 *1.54*	0.07 *0.52*	0.08 *0.44*	0.02 *0.11*	0.11 *0.61*	0.04 *0.25*
Terpinolene																			
Linalool	0.02 *0.14*	0.09 *0.61*	0.04 *0.23*	0.20 *1.00*	0.08 *0.36*		0.09 *0.53*		0.04 *0.17*	0.05 *0.25*	0.06 *0.29*	0.03 *0.16*	0.18 *1.00*	0.07 *0.44*	0.02 *0.18*	0.03 *0.15*		0.06 *0.32*	0.02 *0.12*
Terpineol																			
Geraniol																			
β-Caryophyllene	0.08 *0.60*	0.20 *1.37*	0.30 *1.62*	0.29 *1.42*	0.09 *0.42*	0.45 *1.99*	0.32 *1.87*	0.20 *1.61*	0.29 *1.44*	0.28 *1.42*	0.30 *1.57*	0.13 *0.74*	0.22 *1.21*	0.17 *1.09*	0.09 *0.64*	0.15 *0.86*	0.06 *0.33*	0.17 *0.93*	0.05 *0.34*
Humulene	0.03 *0.24*	0.05 *0.37*	0.13 *0.72*	0.13 *0.62*	0.03 *0.13*	0.12 *0.53*	0.08 *0.45*	0.06 *0.44*	0.12 *0.61*	0.12 0.62	0.13 *0.69*	0.05 *0.25*	0.10 *0.54*	0.05 *0.30*	0.03 *0.21*	0.05 *0.29*		0.07 *0.36*	0.02 *0.13*
Nerolidol	0.01 *0.10*	0.03 *0.19*	0.03 *0.16*	0.12 *0.60*					0.10 *0.47*	0.06 *0.31*	0.08 *0.42*	0.03 *0.19*	0.09 *0.51*		0.02 *0.18*	0.04 *0.20*	0.03 *0.18*		
Guaiol	0.05 *0.37*	0.07 *0.49*			0.10 *0.45*		0.13 *0.74*					0.04 *0.24*		0.05 *0.34*	0.05 0.36	0.07 *0.40*	0.02 *0.12*	0.05 *0.28*	0.04 *0.24*
Bisabolol	0.17 *1.34*	0.15 *1.05*	0.19 *1.02*	0.03 *0.13*	0.10 *0.48*	0.08 *0.38*	0.22 *1.27*		0.18 *0.88*	0.15 *0.76*	0.15 *0.79*	0.08 *0.42*	0.02 *0.09*	0.11 *0.70*	0.15 *1.15*	0.26 *1.47*	0.06 *0.32*	0.05 *0.29*	0.03 *0.18*
Total MONO ^2^	0.42 *3.25*	0.9 *6.31*	1.04 *5.72*	1.27 *6.31*	1.14 *5.35*	1.62 *7.22*	0.62 *3.61*	1.27 *10.13*	1.27 *6.22*	1.14 *5.77*	1.26 *6.60*	0.96 *5.33*	1.09 *6.02*	0.72 *4.66*	0.47 *3.57*	0.58 *3.17*	0.15 *0.8*	0.5 *2.67*	0.33 *2.13*
Total SESQUI ^3^	0.34 *2.65*	0.5 *3.47*	0.65 *3.52*	0.56 *2.78*	0.32 *1.48*	0.65 *2.90*	0.75 *4.32*	0.26 *2.05*	0.69 *3.40*	0.61 *3.11*	0.66 *3.47*	0.33 *1.83*	0.43 *2.36*	0.38 *2.44*	0.34 *2.54*	0.57 *3.22*	0.17 *0.96*	0.34 *1.86*	0.14 *0.89*
Total terpene content	0.76 5.9	1.40 *9.79*	1.68 *9.24*	1.83 *9.09*	1.43 *6.83*	2.27 *10.11*	1.36 *7.93*	1.52 *12.19*	1.94 *9.63*	1.75 *8.87*	1.91 *10.07*	1.30 *7.16*	1.51 *8.37*	1.11 *7.09*	0.82 *6.11*	1.14 *6.39*	0.32 *2.10*	0.83 *4.53*	0.47 *3.02*
MONO ^1^ out of total terpene content (%)	55.1	64.5	61.9	69.5	78.3	71.4	45.5	83.1	64.7	65.0	65.5	74.4	71.9	65.6	58.4	49.6	54.5	58.9	70.5

^1^ % represents percent weight/weight. ^2^ MONO relates to the sum of monoterpenes and monoterpenoids. ^3^ SESQUI relates to the sum of sesquiterpenes and sesquiterpenoids. Inf = inflorescence, THC = tetrahydrocannabinol, THCA = tetrahydrocannabinolic acid, CBD = cannabidiol, CBDA = cannabidiol acid. Total cannabinoid content was calculated as if all of the cannabinoids were in their decarboxylated form. Blank cells indicate terpene levels which are under reporting limit. α-terpinene, p-cymene, γ-terpinene, iso-pulegol, eucalyptol, borneol and caryophyllene oxide were also assessed and were found to be under quantification limit.

**Table 3 molecules-27-06920-t003:** Cannabinoid and terpene compositions in commercial cannabis oil products in Israel. Data is presented as absolute percentage ^1^ and in the form of milligrams of each terpene per 100 mg total cannabinoid content (*in italic*).

Compound Name	Oil 1	Oil 2	Oil 3	Oil 4	Oil 5	Oil 6	Oil 7	Oil 8	Oil 9	Oil 10	Oil 11	Oil 12
THCA			0.2	0.2		0.2					0.3	0.2
THC	5.3	20.0	15.4	10.1	3.0	10.9	5.1	1.4	1.1	1.0	21.3	21.9
Total THC	5.3	20.0	15.6	10.3	3.0	11.1	5.1	1.4	1.1	1.0	21.6	22.2
CBDA												
CBD	10.4	4.4	3.3	2.2	15.0	10.8	10.4	28.0	28.1	20.3	4.5	3.3
Total CBD	10.4	4.4	3.3	2.2	15.0	10.8	10.4	28.0	28.1	20.3	4.5	3.3
Total quantified cannabinoids	15.7	24.4	18.9	12.5	18.0	21.9	15.5	29.4	29.2	21.3	26.1	25.7
α-Pinene												
Camphene												
Sabinene												
β-Pinene												
Myrcene	0.03 *0.18*	0.04 *0.15*	0.03 *0.16*		0.04 *0.21*	0.04 *0.17*	0.03 *0.20*	0.02 *0.08*	0.04 *0.12*		0.03 *0.13*	0.03 *0.12*
Carene												
Ocimene									0.01 *0.04*			
Limonene		0.03 *0.13*								0.01 *0.03*	0.03 *0.10*	
Terpinolene												
Linalool	0.03 *0.18*	0.07 *0.27*	0.06 *0.29*	0.03 *0.23*	0.03 *0.14*	0.05 *0.23*	0.03 *0.21*	0.03 *0.09*	0.02 *0.06*	0.04 *0.17*	0.08 *0.31*	0.06 *0.21*
Terpineol												
Geraniol												
β-Caryophyllene	0.19 *1.20*	0.41 *1.67*	0.33 *1.77*	0.18 *1.41*	0.13 *0.72*	0.26 *1.19*	0.16 *1.01*	0.07 *0.24*	0.07 *0.24*	0.20 *0.96*	0.43 *1.65*	0.33 *1.28*
α-Humulene	0.05 *0.35*	0.13 *0.53*	0.11 *0.56*	0.06 *0.46*	0.04 *0.24*	0.08 *0.38*	0.05 *0.34*	0.02 *0.09*	0.03 *0.09*	0.06 *0.31*	0.14 *0.52*	0.11 *0.41*
Nerolidol			0.07 *0.36*			0.06 *0.27*		0.01 *0.04*	0.01 *0.03*	0.02 *0.09*	0.07 *0.26*	0.07 *0.26*
Guaiol		0.07 *0.30*	0.06 *0.32*	0.03 *0.23*		0.06 *0.27*	0.02 *0.13*	0.17 *0.59*	0.09 *0.32*	0.11 *0.54*	0.10 *0.38*	0.06 *0.23*
Bisabolol	0.06 *0.41*	0.18 *0.74*	0.15 *0.79*	0.09 *0.76*	0.06 *0.34*	0.13 *0.60*	0.06 *0.39*	0.15 *0.52*	0.03 *0.11*	0.16 *0.75*	0.24 *0.93*	0.15 *0.57*
Total MONO ^2^	0.06 *0.36*	0.14 *0.55*	0.09 *0.45*	0.03 *0.23*	0.07 *0.35*	0.09 *0.40*	0.06 *0.41*	0.05 *0.17*	0.07 *0.25*	0.05 *0.27*	0.14 *0.44*	0.09 *0.33*
Total SESQUI ^3^	0.3 *1.96*	0.79 *3.24*	0.72 *3.80*	0.36 *2.86*	0.23 *1.30*	0.59 *2.71*	0.29 *1.87*	0.42 *1.48*	0.23 *0.79*	0.55 *2.65*	0.98 *3.74*	0.72 *2.75*
Total Terpene content	0.36 *2.32*	0.93 *3.79*	0.81 *4.25*	0.39 *3.09*	0.30 *1.65*	0.68 *3.11*	0.35 *2.28*	0.47 *1.65*	0.30 *1.04*	0.60 *2.92*	1.12 *4.18*	0.81 *3.08*
MONO ^1^ out of total terpene content (%)	16.7	15.0	11.1	7.7	23.3	13.3	16.7	10.6	24.1	8.3	12.5	11.1

^1^ % represents percent weight/weight. ^2^ MONO relates to the sum of monoterpenes and monoterpenoids. ^3^ SESQUI relates to the sum of sesquiterpenes and sesquiterpenoids. THC = tetrahydrocannabinol, THCA = tetrahydrocannabinolic acid, CBD = cannabidiol, CBDA = cannabidiol acid. Total cannabinoid content was calculated as if all of the cannabinoids were in their decarboxylated form. Blank cells indicate terpene levels which are under reporting limit. α-terpinene, p-cymene, γ-terpinene, iso-pulegol, eucalyptol, borneol and caryophyllene oxide were also assessed and were found to be under quantification limit.

**Table 4 molecules-27-06920-t004:** Cannabinoid and terpene compositions in various inflorescences and in decarboxylated extracts produced from them. Data is presented as absolute percentage ^1^ and in the form of milligrams of each terpene per 100 mg total cannabinoid content (*in italic*).

	Inf 1	Ext1	Inf 2	Ext 2	Inf 3	Ext 3	Inf 4	Ext 4	Inf 5	Ext 5	Inf 6	Ext 6	Inf 7	Ext 7	Inf 8	Ext 8	Inf 9	Ext 9	Inf 10	Ext 10
THCA	7.8	1.6	12.9	0.6	13.7	5.3	12.8	2.8	10.3	2.6	13.9	4.8	16.8	0.9	17.1	2.2	16.7	0.9	20.3	0.6
THC	1.2	58.4	1.6	61	1	54.2	0.6	57.3	1.1	59	0.7	56.1	0.8	62.3	0.6	57.2	0.6	69.9	0.7	62.4
Total THC	8	59.8	12.9	61.5	13	58.8	11.8	59.8	10.1	61.3	12.9	60.3	15.5	63	15.5	59.1	15.3	70.4	18.5	62.9
CBDA																				
CBD																				
Total CBD	<0.1	<0.1	<0.1	<0.1	<0.1	<0.1	<0.1	<0.1	<0.1	<0.1	<0.1	<0.1	<0.1	<0.1	<0.1	<0.1	<0.1	<0.1	<0.1	<0.1
Total quantified cannabinoids	8	59.8	12.9	61.5	13	58.8	11.8	59.8	10.1	61.3	12.9	60.3	15.5	63.0	15.5	59.1	15.3	70.4	18.5	62.9
α-Pinene	0.1 *1.25*		0.15 *1.12*				0.07 *0.57*		0.02 *0.16*		0.18 *1.4*		0.03 *0.1**5*		0.02 *0.**14*		0.83 *5.43*		0.07 *0.**36*	
Camphene																				
Sabinene																				
β-Pinene	0.04 *0.51*		0.06 *0.47*		0.02 *0.1*		0.02 *0.16*		0.03 *0.26*		0.07 *0.5*		0.04 *0.24*		0.04 *0.27*	0.03 *0.04*	0.19 *1.21*		0.13 *0.64*	
Myrcene	0.03 *0.38*		0.06 *0.44*		0.03 *0.2*		0.06 *0.53*	0.01 *0.02*	0.14 *1.37*	0.02 *0.03*	0.08 *0.6*	0.03 *0.05*	0.37 *2.3*		0.62 *4.1*	0.03 *0.04*	0.30 *1.94*	0.04 *0.05*	0.39 *2.1*	
Carene																				
Ocimene											0.01 *0.1*									
Limonene	0.03 *0.36*		0.05 *0.38*		0.04 *0.3*	0.03 *0.06*		0.02 *0.04*	0.11 *1.13*	0.03 *0.06*	0.06 *0.5*	0.05 *0.09*	0.11 *0.7*		0.14 *0.93*	0.02 *0.03*	0.10 *0.66*	0.05 *0.07*	0.52 *2.78*	0.10 *0.14*
Terpinolene																0.05 *0.0**8*				
Linalool		0.12 *0.2*		0.15 *0.25*	0.02 *0.2*	0.12 *0.21*	0.02 *0.15*	0.1 *0.16*	0.06 *0.55*	0.32 *0.52*	0.02 *0.1*	0.1 *0.17*	0.06 *0.**3**9*	0.14 *0.21*	0.05 *0.35*	0.12 *0.21*	0.04 *0.24*	0.12 *0.16*	0.16 *0.84*	0.35 *0.56*
RT 13.0 *	0.03 *0.33*	0.22 *0.37*	0.04 *0.34*	0.2 *0.33*	0.03 *0.2*	0.12 *0.21*		0.03 *0.06*	0.04 *0.38*	0.22 *0.37*	0.04 *0.3*	0.19 *0.31*	0.04 *0.28*							
RT 14.7 *	0.01 *0.18*		0.02 *0.19*	0.11 *0.18*		0.06 *0.1*			0.03 *0.26*	0.16 *0.26*	0.02 *0.2*	0.1 *0.17*								
Terpineol		0.06 *0.1*		0.05 *0.08*								0.06 *0.1*	0.02 *0.15*	0.07 *0.10*	0.02 *0.14*	0.06 *0.10*	0.02 *0.15*	0.01 *0.01*	0.07 *0.4*	0.19 *0.31*
Geraniol																				
RT19.1 *				0.03 *0.06*		0.07 *0.12*		0.04 *0.07*		0.02 *0.04*		0.02 *0.03*	0.04 *0.26*	0.09 *0.13*		0.04 *0.07*				
β-Caryophyllene	0.06 *0.77*	0.57 *0.98*	0.11 *0.82*	0.67 *1.09*	0.14 *1.1*	0.64 *1.09*	0.13 *1.04*	0.6 *1*	0.13 *1.27*	0.68 *1.11*	0.08 *0.6*	0.37 *0.61*	0.45 *3.0*	0.96 *1.48*	0.44 *2.88*	0.99 *1.66*	0.40 *2.62*	1.26 *1.71*	0.57 *3.0*	1.08 *1.73*
α -Humulene	0.02 *0.26*	0.17 *0.28*	0.04 *0.28*	0.2 *0.33*	0.03 *0.2*	0.17 *0.29*	0.04 *0.3*	0.19 *0.31*	0.04 *0.36*	0.2 *0.33*	0.03 *0.2*	0.13 *0.22*	0.14 *0.90*	0.32 *0.5*	0.13 *0.85*	0.30 *0.51*	0.08 *0.51*	0.26 *0.36*	0.14 *0.78*	0.27 *0.43*
RT 20.2 *		0.09 *0.15*		0.1 *0.16*		0.1 *0.3*		0.02 *0.04*		0.04 0.07	0.01 *0.1*	0.04 *0.07*	0.22 *1.4*	0.49 *0.75*				0.05 *0.07*		0.03 *0.08*
RT 20.4 *		0.07 0.11		0.12 0.2	0.04 *0.3*	0.1 *0.3*	0.01 *0.1*	0.06 *0.1*	0.01 *0.13*	0.08 *0.13*		0.02 *0.03*	0.04 *0.25*	0.08 *0.13*			0.04 *0.24*	0.12 *0.17*	0.07 *0.40*	0.17 *0.27*
RT 20.7 *	0.01 *0.15*	0.08 *0.14*	0.02 *0.16*	0.1 *0.16*		0.06 *0.1*	0.02 *0.14*	0.09 *0.15*	0.01 *0.09*	0.06 *0.09*		0.07 *0.12*			0.02 *0.16*	0.06 *0.1*	0.02 *0.13*	0.06 *0.08*	0.01 *0.10*	0.06 *0.09*
RT 20.8 *		0.03 0.06		0.04 0.06		0.07 *0.12*	0.01 *0.08*	0.05 *0.08*	0.01 *0.09*	0.05 *0.08*	0.01	0.02 *0.04*	0.03 *0.21*		0.02 *0.16*	0.12 *0.20*	0.01 *0.07*	0.06 *0.09*	0.02 *0.10*	0.06 *0.09*
RT 20.9 *	0.02 *0.31*	0.23 *0.38*	0.04 *0.34*	0.26 *0.42*	0.04 *0.3*	0.18 *0.3*	0.06 *0.48*	0.34 *0.57*	0.06 *0.6*	0.36 *0.58*	0.04 *0.3*	0.16 *0.2*	0.05 *0.32*		0.05 *0.33*			0.18 *0.24*	0.02 *0.14*	
RT 21.0 *	0.02 *0.29*	0.19 *0.32*	0.04 *0.29*	0.24 *0.39*	0.03 *0.3*	0.12 *0.21*	0.06 *0.48*	0.35 *0.59*	0.05 *0.52*	0.31 *0.51*	0.04 *0.3*	0.15 *0.25*	0.23 *1.49*							
Nerolidol		0.05 *0.09*		0.08 *0.12*				0.02 *0.03*	0.02 *0.19*	0.12 *0.2*	0.01 *0.1*	0.06 *0.1*			0.03 *0.23*		0.04 *0.24*			
Guaiol	0.03 *0.38*	0.24 *0.42*	0.05 *0.37*	0.27 *0.44*	0.02 *0.2*	0.16 *0.27*			0.06 *0.57*	0.36 *0.58*	0.05 *0.4*	0.26 *0.43*								
Eudesmol *	0.03 *0.41*	0.3 *0.51*	0.05 *0.4*	0.26 *0.43*	0.04 *0.3*	0.16 *0.27*	0.01 *0.08*	0.03 *0.04*	0.06 *0.61*	0.38 *0.62*	0.05 *0.4*	0.26 *0.43*	0.02 *0.14*		0.02 *0.11*		0.02 *0.16*	0.07 *0.10*	0.01 *0.06*	
Bisabolol	0.12 *1.5*	0.84 *1.42*	0.18 *1.38*	0.96 *1.57*	0.11 *0.8*	0.65 *1.11*	0.08 *0.63*	0.4 *0.67*	0.12 *1.24*	0.78 *1.27*	0.17 *1.3*	0.88 *1.47*	0.10 *0.63*	0.25 *0.38*	0.07 *0.45*	0.21 *0.34*	0.03 *0.20*	0.12 *0.16*	0.07 *0.38*	0.15 *0.24*
Total Terpene content	0.55 *7.18*	3.26 *5.43*	0.9 *7.1*	3.84 *6.2*	0.58 *4.5*	2.81 *5.1*	0.59 *4.74*	2.36 *3.93*	1.0 *9.78*	4.2 *6.85*	1.1 *7.4*	2.97 *4.94*	2.0 *12.7*	2.39 *3.72*	1.67 *11.06*	2.11 *3.35*	2.12 *13.66*	2.4 *3.66*	2.25 *12.0*	2.46 *3.63*
Monoterpenes out of total terpenes content (%)	43	12	42	14	23	14	29	9	42	18	50	19	34	13	53	16	69	9	59	20

^1^ % represents percent weight/weight. Inf = inflorescence, THC = tetrahydrocannabinol, THCA = tetrahydrocannabinolic acid, CBD = cannabidiol, CBDA = cannabidiol acid. Total cannabinoid content was calculated as if all of the cannabinoids were in their decarboxylated form. Blank cells indicate terpene levels which are under reporting limit. α-terpinene, p-cymene, γ-terpinene, iso-pulegol, eucalyptol, borneol and caryophyllene oxide were also assessed and were found to be under quantification limit. * Due to lack in analytical standards, content was estimated by calculating terpene’s area from α-humulene response factor. Retention Time (RT) of identified terpenes (min): α-Pinene (6.5); Camphene (7); Sabinene (7.3), Myrcene (7.4), β-Pinene (7.5), Carene (8.3), α-Terpinene (8.5), Ocimene (8.7), p-Cymene (8.9), Limonene (8.8), γ-Terpinene (9.7), Eucalyptol (9.2), Terpinolene (10.6), Linalool (11.5), Iso-Pulegol (13.2), Borneol (14), Terpineol (14.2), Geraniol (15.5), β-Caryophyllene (18.9), α-Humulene (19.6), Nerolidol (21.2), Guaiol (22.2), Caryophyllene oxide (22.3), Eudesmol (22.8), Bisabolol (23.3).

## Data Availability

The data presented in this study are available on request from the corresponding authors.

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
