# Peer review of "Optimal Treatment with Cannabis Extracts Formulations Is Gained via Knowledge of Their Terpene Content and via Enrichment with Specifically Selected Monoterpenes and Monoterpenoids"

_molecules, 2022, doi:10.3390/molecules27206920_

Round 1

Reviewer 1 Report

I cannot recommend acceptance of the paper in the present state. This might be possible only after the authors would have submitted a completely revised version.

Line 13: Please add number of samples.

Line 18: Please add results (in numbers).

Please define terms ‘’medical cannabis’’, ‘’cannabis extract’’, ‘’commercial inflorescences’’, ‘’chemovars‘’,  and ‘’cannabis oil’’ in Introduction.

Table 1: Please arrange terpene class and molecular formula in one column.

Line 77: Please add exact aims of the study and what was measured.  

Line 319: Please add dilution ratio and number of samples.

Line 320: Pure standards or in solvent? What was the mass (or concentration in case of solution)?

Line 330: How did you prepare inflorescences for analysis?

Line 331: Please add the reference.

Line 336: Please define ‘’cannabis cake’’.

Line 345: Please add the reference for the HPLC method. Please add mobile phase and volume of the injected sample. Please add manufacturer of the column.

Line 351: Please add exact concentration range.

Line 352: LOD and LOQ: Please use different unit: % of what?

Line 356: Please add manufacturer of the column.

Lines 359-360: Please do not use ppm unit (ppm of what?)

Line 362: FID cannot give structural data. How can you be sure on exact compound based only on retention time? You need MS for that.

I propose to combine Tables 2 and 3. The same comment for the Tables 4 and 5.

All results should be in past tense.

Line 143: Authors should comment pharmacokinetics after oral ingestion. What about first pass metabolism? What is the % of absorption after oral intake?

Generally, tables should be organized differently to minimize number of data, there are lot of duplicate data in tables.

Lines 247-258 and 261-265: Please move in the Introduction.

The authors use some acronyms without their previous explanation, e.g. FID

All abbreviations in tables should be explained in the footnote.

Figure S1 and S2: Concentration of compounds in analytical standard is missing.

Reviewer 2 Report

I enjoyed reading this manuscript very much.  While a lot of the findings and conclusions are obvious to experienced botanical chemists, this article goes a long way to dispel a number of important mis-understandings amongst the less-scientific players & enthusiasts in the medicinal cannabis industry/field.

The following were pleasing to see:

- Comments about "whole plant" & "full spectrum" extracts

- proper use of the terms terpene, terpenoid, sesquiterpene, sesquiterpenoid

- quantification of terpenes relative to cannabinoid content (mg terpene/100 mg cannabinoid)

- use of the term chemovar

A few suggestions for improvement:

- it is worth mentioning that; one big reason why patients don't like consuming cannabis oils is apparently the taste, vaporizers that don't burn the dried inflorescences might be a good alternative to smoking, wafers and lozenges are alternatives to oils, the differences in therapeutic effects of products with similar cannabinoid contents might result from differences in bioactive components other than terpenes eg flavonoids

- I think the English convention is "cannabinoid content" and "terpene content" not cannabinoids content and terpenes content

- where quantification is reported, ie Tables 2-7, an indication of the uncertainties in the reported numbers should be provided - in the footnotes would be sufficient eg "uncertainties were within 5% of the reported value" or something similar

- does "olive-diluted" mean olive oil diluted?

- line 175 should "lose" be "loss"?

- Figure 3; there seems to be some information missing/incorrect for the x-axis labels. The 3 groupings have the same labels but they clearly represent the analysis of different types of samples, esp. when one reads the figure label.  Also mention of MONO & SESQUI in the label gives a hint to what labels might be missing from the figure.

- the various therapeutic and biological properties of terpenes mentioned on page 13 are largely a matter of conjecture and have not been proven in rigorous clinical trials, but I note that the term "have been described as having ..." is probably suitably non committal.  However, in a rigorous scientific publication, one should be careful not to state anecdotal experiences from the recreational community as fact - see particularly line 264

Overall, I am happy to recommend publication subject to the minor amendments/additions mentioned above.

Round 2

Reviewer 1 Report

All the recommendations and issues raised have been answered and amended accordingly.